# Dynamics of PHA-Accumulating Bacterial Communities Fed with Lipid-Rich Liquid Effluents from Fish-Canning Industries

**DOI:** 10.3390/polym14071396

**Published:** 2022-03-29

**Authors:** David Correa-Galeote, Lucia Argiz, Angeles Val del Rio, Anuska Mosquera-Corral, Belen Juarez-Jimenez, Jesus Gonzalez-Lopez, Belen Rodelas

**Affiliations:** 1Microbiology Department, Faculty of Pharmacy, University of Granada, 18011 Granada, Spain; belenjj@ugr.es (B.J.-J.); jgl@ugr.es (J.G.-L.); mrodelas@ugr.es (B.R.); 2Microbiology and Environmental Technology Section, Water Institute, University of Granada, 18011 Granada, Spain; 3Department of Chemical Engineering, CRETUS Institute, University of Santiago de Compostela, 15782 Santiago de Compostela, Spain; luciaargiz.montes@usc.es (L.A.); mangeles.val@usc.es (A.V.d.R.); anuska.mosquera@usc.es (A.M.-C.)

**Keywords:** bioplastics, PHA accumulation, fish-canning WWTP, bacterial diversity, network correlation analysis, mixed microbial cultures, sequencing batch reactors

## Abstract

The biosynthesis of polyhydroxyalkanoates (PHAs) from industrial wastes by mixed microbial cultures (MMCs) enriched in PHA-accumulating bacteria is a promising technology to replace petroleum-based plastics. However, the populations’ dynamics in the PHA-accumulating MMCs are not well known. Therefore, the main objective of this study was to address the shifts in the size and structure of the bacterial communities in two lab-scale sequencing batch reactors (SBRs) fed with fish-canning effluents and operated under non-saline (SBR-N, 0.5 g NaCl/L) or saline (SBR-S, 10 g NaCl/L) conditions, by using a combination of quantitative PCR and Illumina sequencing of bacterial 16S rRNA genes. A double growth limitation (DGL) strategy, in which nitrogen availability was limited and uncoupled to carbon addition, strongly modulated the relative abundances of the PHA-accumulating bacteria, leading to an increase in the accumulation of PHAs, independently of the saline conditions (average 9.04 wt% and 11.69 wt%, maximum yields 22.03 wt% and 26.33% SBR-N and SBR-S, respectively). On the other hand, no correlations were found among the PHAs accumulation yields and the absolute abundances of total *Bacteria*, which decreased through time in the SBR-N and did not present statistical differences in the SBR-S. *Acinetobacter*, *Calothrix*, *Dyella*, *Flavobacterium*, *Novosphingobium*, *Qipengyuania*, and *Tsukamurella* were key PHA-accumulating genera in both SBRs under the DGL strategy, which was revealed as a successful tool to obtain a PHA-enriched MMC using fish-canning effluents.

## 1. Introduction

Global climate change is currently a major environmental concern due to the high increase of CO_2_ emissions in the last years [1]. Among the different threats, burning fossil fuels is the primary source of CO_2_ emission worldwide, accounting for 77% [2]. In this sense, the manufacturing of conventional plastics (polyethylene, polypropylene, polystyrene, polyester, nylon, acrylic, or polyamide) involves the use of crude oil and natural gas. Along with climate change, the accumulation of traditional petroleum-based plastics produces severe damage to the environment [3]. Worldwide production of plastic polymers continuously rose from 2 million tons in 1950 to around 381 million tons in 2015 [2]. Similarly, the total plastic waste accumulation was estimated to be about 25,000 million tons [4]. Hence, the urgent need to develop biodegradable, compostable, and environmentally friendly substitutes for conventional plastics is widely recognized to meet the continuously increasing demand for materials, guaranteeing future economic and ecological security [5].

Bioplastic production is one of the most promising strategies to cope with the global fossil fuel crisis [1]; however, it currently accounts for only 1% of plastics produced annually [6]. Bioplastics, having similar functions to petroleum-based plastics, can be made from eco-friendly feedstock without depleting natural resources [7]. Furthermore, after their use, they can be turned into noncontaminant substances via composting [8]. Among the biodegradable plastics, polyhydroxyalkanoates (PHAs) display several advantages over other materials [1], attracting considerable attention for their wide range of applications in industrial and medical fields [9]. PHAs are biopolyesters of alkanoic acids, containing a hydroxyl group and at least one functional group attached to the carboxyl group [10]. PHAs can be synthesized and accumulated in cells under unfavorable growth conditions by a wide range of bacteria, such as *Azospirillum* [11], *Bacillus* [12], *Calothrix* [13], *Cupriavidus* [14], or *Pseudomonas* [15]. However, it is mandatory to develop a high-yield and cost-effective process to manufacture PHAs, in order to make them competitive in the market against fossil-fuel-derived plastics [16].

The imperative for PHAs biosynthesis by using an MMC is by achieving an adequate enrichment of PHA-accumulating bacteria [7]. Among the different approaches developed, the feast–famine (F/F) strategy has been characterized as the most effective [17,18]. The F/F regime consists of the initial subjection of the bacterial culture to an excess of carbon source (feast), followed by a carbon deficiency (famine), under aerobic conditions [19]. According to Queirós et al. [20], the application of alternating excess and limitation of external carbon substrates provides a selective pressure favorable to PHA-storing bacteria, allowing for their enrichment. Throughout the initial feast phase, PHA-accumulating bacteria can store these polymers that are subsequently used as an internal carbon source during the lack of external substrate in the famine phase, making this trait a competitive advantage over the other microorganisms [18]. Besides, the MMC needs to be robust and resilient to adapt to the physicochemical and operational changes that the PHA-accumulating process usually involves when wastes are used as substrate [17].

Several works have reported the conversion of different carbon sources into PHAs using MMC, most of them emphasizing the influence of operating conditions and the development of new PHAs purification methods [21,22,23,24,25,26]. In addition, different studies have addressed PHAs accumulation in MMCs using industrial wastes as an alternative carbon feedstock for the biological production of PHAs [17,27,28]. In this sense, the use of wastes generated in the food processing industry has been proposed because they often contain high levels of fat, oil, and grease (FOG) [29]. In particular, lipid-rich liquid effluents generated from fish-canning companies stand out for their inexpensiveness, wide availability, and high contents of sugar and oils, which make them advantageous over other substrates, such as edible plants oils [30]. The straight valorization of these wastes provides a significant advance for the implementation of a circular economy in fish-canning industries, besides preventing environmental damage [31]. However, PHAs production from FOG-enriched streams often requires pretreatment operations, which would increase production costs [32]. Furthermore, fish-canning wastes are characterized by high contents of salt and organic matter, which can affect microbial communities in their composition and robustness [33,34,35]. Therefore, an optimization of the process is required. In this sense, Argiz et al. [36,37] have recently demonstrated the feasibility of valorizing FOG-enriched fish-canning waste into PHAs in lab-scale MMC systems, under low or high concentrations of NaCl (0.5 or 10 g NaCl/L, respectively).

Generally considered, the different alternatives tested for PHA-production from industrial wastes by MMCs have not yet achieved optimized yields, compared to those obtained when pure carbon sources [38] and pure bacterial cultures were used [20]. To overcome such a problem, better biotechnologies imposing the proper selective pressures to achieve a robust enrichment of the complex MMCs and enable higher PHAs storage capacities are needed [1]. Therefore, several questions regarding the size and structure of the bacterial communities in PHA-accumulating MMCs need to be answered, mainly regarding the successional patterns of the key genera involved in PHA-biosynthesis. Similarly, the relationships between the populations’ dynamics of the MMCs, the PHAs accumulation yields, and the different operational variables are not well known. In particular, the effect of the high salinity often found in fish-canning effluents over the establishment of a PHA-accumulating MMC should be further investigated, since this has been seldom addressed in the previous literature [36,37,39,40]. Hence, the main objective of the present study was to analyze the shifts of the size and structure of the bacterial communities of two MMCs from PHA-accumulating processes using FOG waste as a substrate, by using quantitative PCR (qPCR) and Illumina sequencing, respectively. Furthermore, the relationships among the bacterial populations’ dynamics of the MMCs, the PHA accumulation yields, and the changes of operational variables of the SBRs (particularly the NaCl concentration), were explored.

## 2. Materials and Methods

### 2.1. SBRs Set-Up and Operation for the PHA-Enrichment Strategy

Two lab-scale (4 L) sequencing batch reactors (SBRs) were operated for the enrichment of PHA-accumulating MMCs under non-saline (SBR-N, 0.5 g NaCl/L) and saline (SBR-S, 10 g NaCl/L) conditions. Each reactor was operated with a 12 h cycle at 30 ± 3 °C under a F/F regime with continuous aeration. pH was controlled by the addition of a NaHCO_3_ solution. Activated sludge (AS) from an urban wastewater treatment plant (WWTP) was used as inoculum for the SBR-N, and an AS from a WWTP treating fish-canning waste was employed for the SBR-S. The substrate used for PHAs’ accumulation was a residual FOG-enriched stream from the cooking water of a fish-canning industry placed in Galicia (NW Spain), and the volume of substrate added in each cycle was 2 mL, equivalent to 114.5 Cmmol/cycle. Half of the reactor volume was exchanged from the SBR at the end of the cycle, resulting in hydraulic (HRT) and solid (SRT) retention times of 24 h. Besides, depending on the feeding strategy, 2 L of non-saline and saline dilution water were supplied at different times to the SBR-N and SBR-S, respectively.

SBR-N and SBR-S were operated for 331 and 122 days, subdivided into four comparable operational periods (NS, NI, NII, and NIII for SBR-N; and SS, SI, SII, and SIII for SBR-S, Appendix A). Briefly, the start-up (NS and SS, for SBR-N and SBR-S, respectively) was operated under the F/F regime (conventional aerobic dynamic feeding strategy (ADF)) until reaching the steady-state operation. In the following period, for the maturation phase (NI and SI, for SBR-N and SBR-S, respectively), the same F/F strategy was employed. In the third operational period (NII and SII, for SBR-N and SBR-S, respectively), a double growth limitation (DGL) strategy was imposed, in which carbon (added in the feast phase) and nitrogen (added in the famine phase) feedings were uncoupled. Thus, nitrogen availability was limited during the feast phase, establishing an additional selective pressure advantageous for PHA-storing microorganisms [41,42]. In the last operational period (NIII and SIII, for SBR-N and SBR-S, respectively), the extent of the impact of pH control on the PHAs’ accumulation yields was tested by operating both SBRs under the same DGL strategy combined with a reduction of the NaHCO_3_ supply during the famine phase. Details concerning the cycle configurations can be consulted in Argiz et al. [36,37].

### 2.2. Chemical Determinations

The pH of the feeding and at the end of the cycle were monitored by a pH meter (GLP22, Crison, Alella, Spain). Volatile suspended solids (VSS) were determined according to the standard methods [43]. Samples filtered through a 0.45 µm pore size cellulose-ester membrane (Advantec, Tokyo, Japan) were employed to determine total organic carbon (TOC) and total nitrogen (TN) by using a TOC-L analyzer plus the TNM-module (Shimadzu, Kyoto, Japan). NaCl concentrations were quantified in bulk samples using an advanced compact ion chromatograph (Metrohm, Herisau, Switzerland).

Quantification of PHAs was carried out in lyophilized biomass samples by gas chromatography, according to Argiz et al. [36]. A copolymer containing 88% hydroxybutyrate and 12% hydroxyvalerate (Merck KGaA, Darmstadt, Germany) was used as a calibration standard. The PHAs’ contents of the biomasses were expressed as the percentage of the VSS dry weight.

### 2.3. Microbial Analysis

Samples (9–332 mL) from both SBRs were centrifuged at 14,000 rpm for 1 min, supernatants were discarded and, finally, the resulting biomasses were frozen at −20 °C until DNA extraction. Two independent replicates were used for each DNA extraction, utilizing the FastDNA Spin kit for soil in the FastPrep-24 system (MP Biomedicals, Irvine, CA, USA) according to the manufacturer’s instructions.

Quantification of small-subunit rRNA and functional marker genes of *Bacteria* was performed by qPCR on a QuantStudio-3 Real-Time PCR instrument (Applied Biosystems, Waltham, MA, USA). Briefly, the bacterial 16S rRNA gene was used as a proxy for the total size of the *Bacteria* community. Quantification of 16S RNA genes of genus *Candidatus* Microthrix (hereafter referred to as Microthrix) and the Mycolata group (including 17 genera of *Actinobacteria* enriched in mycolic acids, mainly *Gordonia*, *Mycobacterium* and *Nocardia*), were used to assess the abundance of filamentous bacteria. Finally, 16S rRNA genes of *Candidatus* Accumulibacter and *Candidatus* Competibacter were used as a proxy for polyphosphate- (PAOs) and glycogen-accumulating organisms (GAOs), respectively. Primer sequences, reaction mixtures, and cycling conditions are fully described in Correa-Galeote et al. [35].

High-throughput Illumina MiSeq sequencing of the V3-V4 region of the 16S rRNA gene was performed using the primers Pro341F/Pro805R [44] in the facilities of the Institute of Parasitology and Biomedicine “López-Neyra” (IPBLN). Raw data from Illumina sequencing were processed using the Mothur v1.44.1 software, according to the guidelines provided in the Mothur webpage (www.mothur.org/wiki/miseq_sop/, accessed on 12 September 2021). Firstly, the corresponding forward and reverse reads were combined to merge the files; after, contigs were generated. The sequences were subjected to quality filtering and primer trimming, and the whole set of improved unique sequences was aligned (MAFFT algorithm, [45]) using them as a seed random subsample (10,000 sequences) in the Geneious 2021.1.1 software (Biomatters, Auckland, New Zealand). Afterwards, a de-noising analysis and removal of chimeric sequences were also made. The resulting sequences were clustered into OTUs (97% similarity threshold), and only OTUs with an abundance of >0.0001% (8 sequences) were retained for subsequent analysis. Finally, the consensus sequence of each OTU was taxonomically identified using the curated collection of bacterial 16S rRNA sequences of the NCBI (www.ncbi.nlm.nih.gov/refseq/targetedloci/16S_process, accessed on 15 October 2021) through the BLAST tool of the Geneious software. Nucleotide sequences were deposited in the GeneBank repository (accession number SUB11147402). Simpson and Shannon diversity indices were calculated according to Hill et al. [46].

### 2.4. Statistical Analysis

The statistical differences (*p* < 0.05 significance level) among the corresponding samples were explored through the nonparametric Mann–Whitney or Kruskal–Wallis (Conover–Iman post hoc analysis) tests using the XLSTAT v2021.1.1 software (Addinsoft, New York, NY, USA). Corresponding Spearman correlations were determined in XLSTAT and a co-occurrence network of these significant correlations (*p* < 0.05 significance level) was constructed using Gephi v0.9.2 (Gephi Consortium, Paris, France). A heatmap was created using the average clustering method with R studio v2019.3.2 (RStudio, Boston, MA, USA). Finally, a nonmetric multidimensional (NMS) analysis was made using the PC-ORD software (Wild Blueberry Media, Corvallis, OR, USA).

## 3. Results and Discussion

### 3.1. PHAs Accumulation Ability

Appendix A shows the average values of the main physicochemical parameters and the PHAs accumulation capacity in the different operational periods for both enrichment SBRs. Generally considered, a scarce number of significant differences were found for the different physicochemical data. Regarding the PHAs accumulation capacity, it was negligible during the start-up of both SBRs (NS and SS), (average values 0.10 wt% and 0.18%; NS and SS, respectively; and maximum yields 0.97 wt% and 0.36% at days 65 and 10 for NS and SS, respectively). Once stable operational performances were achieved (NI and SI, for SBR-N and SBR-S, respectively), the mean PHAs accumulations were 1.66 wt% (the maximum yield was 2.15 at day 106) for SBR-N, and 2.07 wt% (the maximum yield was 4.39 wt% at day 45) for SBR-S, respectively (Appendix A). Therefore, the application of a single growth limitation strategy (the F/F regime) failed to obtain adequate levels of PHAs accumulation. However, the implementation of the DGL strategy (NII and SII, for SBR-N and SBR-S, respectively) statistically increased the PHAs accumulations in both SBRs (average values 9.04 wt% and 11.69 wt%, NII and SII, respectively; and maximum yields 22.03 wt% and 26.33% at days 247 and 63 for NII and SII, respectively), compared to those found when only the F/F pressure was applied. Finally, the DGL strategy, together with a reduction of the NHCO_3_ supply (NIII and SIII, for SBR-N and SBR-S, respectively), did not result in significant changes in the PHAs yield under the experimental conditions in both SBRs (mean values 7.25 wt% and 8.10 wt%, NIII and SIII, respectively; and maximum yields 21.66 wt% and 12.00% at days 316 and 122 for NIII and SIII, respectively). Therefore, the absence of pH control had no significant effect on the average PHAs accumulation yield. On the other hand, no significant differences in the PHA-accumulating capacity were found between reactors for a given period, indicating that neither a high level of NaCl (10 g/L) nor the inoculum used affected the PHAs accumulation capacity. Besides, the PHAs accumulation rates described here were similar to those reported in other MMCs fed with wastes [23,47].

### 3.2. Quantification of Total Bacterial Populations and Key Functional Groups

The absolute abundances of total *Bacteria* ranged from 3.29 × 10^7^ to 2.34 × 10^14^ and 5.09 × 10^10^ to 5.08 × 10^12^ bacterial 16S rRNA gene copies/L of biomass in the SBR-N and SBR-S, respectively (Figure 1 and Figure 2, Appendix A). Therefore, SBR-N bacterial populations showed stronger variances in their magnitudes (up to 7 orders of magnitude) than those of SBR-S (only to 2 orders of magnitude). The total bacterial populations’ abundances fell within the values previously reported in other lab-scale SBRs [48] and in different full-scale bioreactors [49].

In SBR-N, the largest size of the bacterial communities was found in the NS period, followed by NI, NIII, and NII (Figure 1 and Figure 2, Appendix A). This result agrees with that of previous work [50], which described that bacterial abundances declined after applying a disturbance pressure. However, there were no statistical differences in the total bacteria abundances in SBR-S among the different operational periods, suggesting a higher resilience in this bioreactor in terms of bacterial community size. On the other hand, the total bacteria abundances were statistically similar when comparing the periods of NS vs. SS and NI vs. SI. However, after applying the DGL strategy, the numbers of bacterial 16S rRNA gene copies were significantly higher in the SBR-S in the remaining experimental periods (SII vs. NII and SIII vs. NIII).

The abundances of Microthrix, Mycolata, PAOs, and GAOs are shown in Appendix A. The copies of marker genes for these populations greatly differed among periods for a given reactor, with the exception of the total abundances of Mycolata for the SBR-S and PAOs for the SBR-N, which were not significantly different among periods. Generally considered, these key groups tended to decrease over time, suggesting that the different selective pressures applied reduced their presence. On the other hand, there were a plethora of statistical differences between both reactors for a given period. However, no clear effects of the NaCl or the inoculum source were found on the sizes of these populations. Both Microthrix and the Mycolata group are filamentous bacteria essential for developing flocs by providing the backbone for biomass aggregation [51]. Accumulibacter and Competibacter are well-known PHA-storing microorganisms [52]; however, the abundances of PAO were more often several orders of magnitude lower than total *Bacteria* in the SBR-N and were mostly below the limit of detection in the SBR-S (Appendix A), suggesting that their contribution to PHAs accumulation was negligible.

### 3.3. Bacterial Communities’ Diversity

The total number of high-quality bacterial sequences was 4,729,690 (average sequences per library 67,607 ± 21,022). The OTUs’ abundance distribution and their taxonomic classification are shown in Appendix A. Overall, 3171 different OTUs were found; and the average number of OTUs per library was 494 ± 386. The alpha diversity indices are shown in Appendix A. In SBR-N, the highest numbers of OTUs were found in the NS period (average 822 OTUs, with 1585 OTUs in the initial inoculum), followed by NII (659 OTUs), NI (509 OTUs), and NIII (214 OTUs). In the SBR-S, the average number of OTUs in the initial inoculum (578) was reduced about threefold after 10 days of operation, but there were no more significant changes of the OTUs richness throughout the remaining operational time (average 232 bacterial OTUs). According to Huang et al. [17], the more substantial reduction in the richness found in the SBR-N throughout operation indicated that more OTUs in the initial inoculum was less able to become adapted to the different pressure conditions compared to those in the SBR-S. Also, the steady-state was faster achieved in the SBR-S, suggesting that the bacterial community in its inoculum held out better than the F/F regime applied since the beginning of the operation.

Regarding the alpha diversity indices, the highest and lowest values of the Simpson and Shannon indices were found in periods NIII and SIII. According to Marzorati et al. [53], the DGL strategy under acid conditions resulted in higher functional organizations due to the lower evenness found in those samples. The trends, followed by the alpha diversity indices through the different experimental periods, were in agreement with those previously described in a PHA-accumulating SBR fed with paper factory residues [54]. Besides, no clear differences in the diversity indices were found between the two bioreactors, suggesting that neither higher NaCl concentration nor the inoculum source had side effects on the bacterial communities’ diversity or evenness.

### 3.4. Dynamics of the Bacterial Communities’ Structure

The 3171 OTUs belonged to 31 different phyla, plus the group of unclassified sequences (157 sequences (5 OTUs), Appendix A). Eight out of the 31 phyla had an average relative abundance (RA) >0.5% and were considered the dominant groups. Overall, the bacterial communities were mainly composed of *Proteobacteria* (average RA 49.72%), *Actinobacteria* (18.42%), *Bacteroidetes* (16.67%), *Cyanobacteria* (5.74%), *Chloroflexi* (2.83%), *Firmicutes* (2.36%), *Verrucomicrobia* (1.74%), and *Acidobacteria* (1.06%). The remaining phyla accounted for only 1.74%. The phylum *Proteobacteria* was composed of 5 classes with an RA > 0.5%, which in turn, were also considered dominant groups, which were the following sorted in decreasing order of average RA: *Alphaproteobacteria* (27.53%), *Betaproteobacteria* (11.35%), *Gammaproteobacteria* (9.00%), *Oligoflexia* (0.83%), and *Deltaproteobacteria* (0.71%). In general, the bacterial community structures at the phylum level in both SBRs were similar to those previously reported in other PHA-accumulating SBRs [17,25,54]. Figure 3 and Figure 4 show the bacterial profiles at the phylum level in all samples.

The RAs of these dominant bacterial groups were significantly different between the operational periods for a given SBR, except for *Actinobacteria*, *Chloroflexi*, *Firmicutes*, *Alphaproteobacteria*, and the minority phyla in SBR-N; and for *Acidobacteria*, *Actinobacteria, Chloroflexi*, *Firmicutes*, *Deltaproteobacteria,* and the minority phyla in SBR-S (Appendix A). Similarly, there were several significant differences in the RAs of the dominant groups between the two SBRs for a given operational period, except for *Gammaproteobacteria* (Appendix A). Despite these significant differences, only the RAs of *Betaproteobacteria* were higher in the four periods of SBR-N compared to those in the SBR-S. Therefore, neither a high concentration of NaCl nor the inoculum employed strongly influenced the structure of the bacterial communities at the phylum level.

There were 894 different bacterial genera, and 42 of them were considered dominant (RA > 0.5% of the total sequences), representing around 70% of the total bacterial sequences. These genera sorted in decreasing order of average RAs were: *Gordonia* (*Actinobacteria*, 8.88%), *Qipengyuania* (*Alphaproteobacteria*, 6.19%), *Acidovorax* (*Betaproteobacteria*, 5.65%), *Taibaiella* (*Bacteroidetes*, 4.08%), *Chryseobacterium* (*Bacteroidetes*, 3.91%), *Kryptousia* (*Cyanobacteria*, 3.61%), *Niveispirillum* (*Alphaproteobacteria*, 3.01%), *Azospirillum* (*Alphaproteobacteria*, 3.00%), *Rhizobium* (*Alphaproteobacteria*, 2.16%), *Calothrix* (*Cyanobacteria*, 2.10%), *Pseudomonas* (*Gammaproteobacteria*, 2.04%), *Mycolicibacterium* (*Actinobacteria*, 1.96%), *Corynebacterium* (*Actinobacteria*, 1.90%), *Comamonas* (*Betaproteobacteria*, 1.78%), *Novosphingobium* (*Alphaproteobacteria*, 1.56%), *Flavobacterium* (*Bacteroidetes*, 1.24%), *Gemmobacter* (*Alphaproteobacteria*, 1.08%), *Leifsonia* (*Actinobacteria*, 1.03%), *Alkaliphilus* (*Firmicutes*, 1.00%), *Tsukamurella* (*Actinobacteria*, 0.86%), *Dokdonella* (*Gammaproteobacteria*, 0.84%), *Lysobacter* (*Gammaproteobacteria*, 0.83%), *Bdellovibrio* (*Oligoflexia*, 0.81%), *Roseiflexus* (*Chloroflexi*, 0.79%), *Thermomonas* (*Gammaproteobacteria*, 0.76%), *Devosia* (*Alphaproteobacteria*, 0.75%), *Polynucleobacter* (*Betaproteobacteria*, 0.73%), *Stenotrophomonas* (*Gammaproteobacteria*, 0.69%), *Mucilaginibacter* (*Bacteroidetes*, 0.69%), *Aquihabitans* (*Actinobacteria*, 0.66%), *Melaminivora* (*Betaproteobacteria*, 0.66%), *Aquabacter* (*Alphaproteobacteria*, 0.65%), *Ravibacter* (*Bacteroidetes*, 0.64%), *Ferruginibacter* (*Bacteroidetes*, 0.63%), *Insolitispirillum* (*Alphaproteobacteria*, 0.61%), *Thalassotalea* (*Gammaproteobacteria*, 0.61%), *Oscillochloris* (*Chloroflexi*, 0.59%), *Pandoraea* (*Betaproteobacteria*, 0.58%), *Acinetobacter* (*Gammaproteobacteria*, 0.58%), *Dyella* (*Gammaproteobacteria*, 0.58%), *Fluviicola* (*Bacteroidetes*, 0.52%), and *Elstera* (*Alphaproteobacteria*, 0.43%). In addition, 28.33% of the sequences classified at the genus level had a RA < 0.5% and were included within the minority genera miscellaneous group. The dominant bacterial genera in each sample are depicted in Figure 5 and Figure 6, and Appendix A.

The most abundant genera identified in the initial inoculum of SBR-N were *Roseiflexus* (7.96%), *Gordonia* (5.79%), *Ferruginibacter* (5.40%), and *Oscillochloris* (5.21%). At the end of the NS period (day 65), the most enriched genera were *Acidovorax* (22.33%), *Chryseobacterium* (20.95%), and *Gordonia* (5.98), while *Acidovorax* (18.22%), *Pseudomonas* (8.65%), *Comamonas* (5.96%), and *Mucilaginibacter* (5.39%) displayed the highest RAs at the end of NI (day 120). *Niveispirillum* (19.07%) and *Gordonia* (18.50%) turned dominant by the end of NII (day 247). Finally, *Calothrix* (20.88%), *Gordonia* (18.15%), *Kryptousia* (17.37%), and *Chryseobacterium* (7.72%) had higher-than-average RAs at the end of NIII (day 323). The minority genera comprised most of the bacterial community in the initial inoculum (65.51%), but the RA of this fraction was significantly reduced throughout operation (20.17, 31.43, 32, 21, and 14.70% at the end of the periods NS, NI, NII, and NIII, respectively). In the SBR-S’ inoculum, the most abundant genera identified were *Aquihabitans* (5.03%), *Mycolicibacterium* (3.73%), *Azospirillum* (2.14%), and *Gordonia* (1.59%), being displaced by *Rhizobium* (26.58%), *Stenotrophomonas* (18.04%), and *Gordonia* (8.65%) at the end of the SS period (day 10). *Qipengyuania* (11.46%), *Flavobacterium* (10.13%), *Chryseobacterium* (8.92%), *Gordonia* (8.29%), and *Rhizobium* (7.57%) were enriched by the end of SI (day 45), while *Novosphingobium* (20.52%), *Insolitispirillum* (18.86%), *Rhizobium* (15.62%), and *Gordonia* (8.96%) turned dominant at the end of period SII (day 86). Finally, *Qipengyuania* (57.09%), *Corynebacterium* (17.33%) and *Dokdonella* (11.62%) were the most abundant genera at the end of SIII (day 122). Similar to what was observed in the SBR-N, the percentage of sequences belonging to the group of minority genera decreased gradually through operation, from 86.45% in the initial inoculum to 42.20, 27.89, 19.35 and 4.80% at the end of periods SS, SI, SII, and SIII, respectively.

According to the aforementioned results, broad successions of the dominant genera were found for both SBRs throughout the operational time. This result agrees with Wen et al. [38], which found wide changes in the bacterial communities’ structure in different PHA-accumulating SBRs using crude glycerol as a substrate. In addition, according to Appendix A, after reaching the steady-state (period NS vs. NI and SS vs. SI), 11 dominant genera were enriched in each SBR, while 8 and 2 genera significantly reduced their RAs in SBR-N and SBR-S, respectively. On the other hand, the implementation of the DGL strategy (NII vs. NI and SII vs. SI) resulted in increments of the RAs of 7 genera and the reduction of 15 other genera in the case of SBR-N. Similarly, in SBR-S, 11 genera increased, and 4 decreased their RAs in SII compared to the average RAs of SI. Finally, the acidification of the medium (NII vs. NIII and SII vs. SIII) strongly affected the bacterial communities once again, increasing the RAs of 9 and 5 genera in SBR-N and SBR-S, respectively; and decreasing the RAs of other 13 dominant groups in each SBR. Therefore, the different strategies applied in the operational periods produced strong shifts in the bacterial communities’ structure in both SBRs. In contrast, Morgan-Sagastume et al. [55] found a stable microbial community during a PHA-accumulating operation in a pilot-scale SBR fed with fermented waste sludge. The broad successions of the dominant populations in the bacterial communities reported here could be linked to the opening of new niches, which could be colonized by different PHA-accumulating bacteria, allowing higher PHAs yields, as previously suggested by Huang et al. [17]. Regarding the differences between reactors for a given period, several statistical differences were found (Appendix A), except for the genus *Stenotrophomonas*. Accordingly, *Acidovorax*, *Polynucleobacter*, *Lysobacter*, and *Bdellovibrio* were significantly more abundant in the SBR-N for each period, and only the RAs of *Qipengyuania* were statistically higher in all periods in the SBR-S.

According to Albuquerque et al. [56], a deficiency of nitrogen during the feast phase is required to achieve higher PHAs accumulation rates. In this study, after the implementation of the DGL strategy, several genera of diazotrophic bacteria increased their RAs (NI vs. NII and SI vs. SII): *Azospirillum*, *Calothrix*, *Corynebacterium*, *Insolitispirillum*, *Kryptousia*, *Niveispirillum*, and *Novosphingobium* [57,58,59,60,61,62,63,64]. More specifically, *Azospirillum*, *Calothrix*, and *Niveispirillum* were enriched in SBR-N, and *Corynebacterium*, *Insolitispirillum*, and *Novosphingobium* in SBR-S. It has been previously described that the PHA-accumulating capacity is amply present in N-fixing bacteria [11,13,65]. In addition, PHAs accumulation is enhanced under the deprivation of combined N in different genera of diazotrophs [66,67]; therefore, their enrichment in MMCs could be advantageous to obtain higher polymer yields when a DGL strategy is implemented. In this sense, high molecular weight PHAs were successfully produced by an MMC enriched in a laboratory-scale SBR lacking N feeding [21]. Alongside their role as C and energy storage polymers, PHAs improve the tolerance of prokaryotes to environmental stress factors [68]. Further research is necessary to evaluate the specific contribution of diazotrophic bacteria in PHA-accumulating SBR.

In addition, the occurrence stability indices of the dominant genera were calculated according to Skorupski et al. [69] to determine the global successional patterns in the bacterial communities. Appendix A shows that the majority of the genera within the dominant bacterial community (29 of the 42 dominant groups) were classified as euconstants, taking into account both SBRs (32 and 27 in SBR-N and SBR-S, respectively) [70]. Therefore, despite the differences observed in their RAs, the dominant genera were highly persistent, confirming the robustness and the resilience of the enriched dominant bacterial communities to ensure the PHAs biosynthesis process [71].

Finally, an estimation of the beta diversity of the bacterial communities of both reactors was made at the beginning and end of each period to assess if deterministic processes drove the global microbial community successions. The Venn diagrams (Appendix A) show the number of shared genera between the reactors and their RAs for each experimental period. According to these results, after each of the different strategies was implemented (end of the corresponding period) higher amounts of shared genera between reactors and at higher RAs were found, compared to those at the beginning of the period, except for NIII and SIII. The latter was due to the high ratio of shared genera between both reactors at the beginning of period III (92.03%). Hence, the dynamics of the bacterial communities were driven by deterministic more than stochastic processes, regardless of the inoculum or the NaCl concentration. In addition, the heatmap of the RAs of the dominant genera (Appendix A) showed that the samples were not ordered according to the bioreactor or the experimental period, which lends support to the hypothesis of a deterministic enrichment of the bacterial communities in both SBRs.

### 3.5. Potential PHAs Accumulation Capacities of the Dominant Bacterial Genera

The dominant genera previously described as PHA-accumulating bacteria were *Acidovorax* [72], *Acinetobacter* [73], *Alkaliphilus* [74], *Azospirillum* [11], *Calothrix* [13], *Comamonas* [75], *Corynebacterium* [76], *Gemmobacter* [77], *Leifsonia* [78], *Lysobacter* [79], *Novosphingobium* [80], *Oscillochloris* [81], *Pandoraea* [82], *Polynucleobacter* [83], *Pseudomonas* [84], *Rhizobium* [85], *Roseiflexus* [86], *Stenotrophomonas* [87], and *Tsukamurella* [55]. Moreover, species of the diazotrophic genus *Niveispirillum* were formerly classified within the well-known PHA-accumulating genus *Azospirillum* [88]. In addition, according to entries in the UniProt database (https://www.uniprot.org, accessed on 6 February 2022), species of the genera *Chryseobacteium*, *Dokdonella*, *Dyella*, *Gordonia*, *Qipengyuania*, *Taibaiella*, and *Thermomonas* carry the *phaC* gene, which codifies the key enzyme of the PHA-biosynthesis [89]. Subsequently, these genera were considered potential PHA-accumulators. Finally, there is no scientific evidence of the PHAs accumulation capacities of the genera *Devosia*, *Ferruginibacter*, and *Flavobacterium*; however, they have been earlier described as dominant in other PHA-accumulating SBRs [90,91,92,93].

On the other hand, the genus *Bdellovibrio* (*Oligoflexia*) is an obligate bacterial predator that can catabolize PHAs, reducing their accumulation yields [94]. Nevertheless, Martínez et al. [95] proposed the use of a PHA-depolymerase knockout mutant of *Bdellovibrio* as an alternative and promising tool for the inexpensive, industrial-scale recovery of intracellular products from different Gram-negative prey cultures, compared to the conventional mechanical cell disruption methods.

Therefore, the PHAs accumulation capacity was widespread over the most abundant genera within the dominant phyla in both SBRs. Overall, the profiles of dominant genera agreed with those previously described in other PHA-accumulating SBRs [17,25,54], although some other well-known PHA-accumulating bacteria (*Bacillus*, *Cupriavidus*, *Nocardia*, or *Rhodococcus* [12,14,96]) were not prevalent in the MMCs. Finally, the average RAs of genera that were potentially able to accumulate PHAs were 58.99% and 64.95% in SBR-N and SBR-S, respectively. Remarkably, this average RA was 92.70% in the SIII period, mainly due to the abrupt increase in *Qipengyuania*. Strikingly, *Qipengyuania* is a non-halotolerant genus, first isolated from a Tibetan permafrost [97].

Considering the use of a lipid-rich waste as a substrate for the PHAs accumulation, it is necessary for the hydrolysis of triglycerides into free fatty acids before their subsequent transformation to PHAs [98]. The bacteria that were potentially able to accumulate PHAs with a lipolytic activity were *Acidovorax* [99], *Acinetobacter* [100], *Azospirillum* [101], *Chryseobacterium* [102], *Comamonas* [103], *Corynebacterium* [104], *Devosia* [105], *Dokdonella* [106], *Dyella* [107], *Ferruginibacter* [108], *Flavobacterium* [109], *Gemmobacter* [77], *Gordonia* [110], *Leifsonia* [111], *Lysobacter* [112], *Niveispirillum* [113], *Novosphingobium* [114], *Pandoraea* [115], *Polynucleobacter* [83], *Pseudomonas* [98], *Qipengyuania* [116], *Rhizobium* [117], *Stenotrophomonas* [118], *Taibaiella* [119], *Thermomonas* [120], and *Tsukamurella* [121].

According to these results, the operational strategy applied in both SBRs resulted in the implementation of well-established PHA-accumulating communities with ample lipolytic activity, despite the observed turnover of community composition throughout the different operational periods.

### 3.6. Network Correlations within the Dominant Bacterial Communities

To evaluate the interactions between the main bacterial genera in SBR-N and SBR-S, the corresponding co-occurrence networks were constructed based on the significant Spearman’s rank correlation coefficients (ρ > 0.4) of their RAs (Figure 7; Appendix A). The resulting networks were composed of 42 nodes and 634 edges (36.82% of the potential correlations) for SBR-N, and 41 and 68 edges (34.64% of the potential correlations) for SBR-S. Subsequently, the enriched bacterial communities were highly imbricated, showing a plethora of synergistic interrelationships among the different dominant genera. The bacterial communities with strong levels of interactions among their members are better adapted to changes in the environment [122], such as those inflicted by the different selective strategies applied in the SBRs. However, the topology of the networks was strongly different, depending on the bioreactor. For the SBR-N, the genera with a high level of positive syntrophic interactions were *Devosia*, *Gemmobacter*, *Melaminivora*, and *Pseudomonas*, and the genera more negatively correlated were *Dyella*, *Pandoraea*, *Kryptousia*, and *Niveispirillum* (Appendix A). On the other hand, for the SBR-S, the more positively connected genera were *Alkaliphilus*, *Devosia*, *Aquabacter*, and *Acidovorax*, and the genera with a higher number of negative correlations were *Calothrix*, *Thalassotalea*, and *Azospirillum* (Appendix A). All the keystone genera contribute to maintaining the bacterial communities’ structure under different perturbations through time [123]. Except for *Melaminivora*, *Kryptousia*, and *Aquabacter*, the remaining dominant genera have potential PHAs accumulation capacities, highlighting their essential role in the resilience of the enriched PHA-accumulating community.

### 3.7. Influence of the Bacterial Communities on the PHAs Accumulation Yield

An NMS analysis, which was based upon the RAs of the 42 dominant genera plus the group of minority genera, was carried out; also, the operational variables and the PHA-accumulating yields were linked to the NMS, aiming to deeper analyze their relationships with the dynamics of the bacterial community structure. As a result, the NMS biplot (Figure 8) shows that the samples retrieved from SBR-N were more similar among them than the samples taken from SBR-S, displaying a narrower distribution on the biplot.

Moreover, the different enrichment strategies applied were driving forces of the shifts experienced by the bacterial communities’ structure in both SBRs, because the different samples were ordinated according to their corresponding operational periods. Remarkably, these enrichment strategies had abrupt impacts on the bacterial community structure of the SBR-S, resulting in the turnover of several dominant genera through the operational time.

The bacterial genera that were potentially able to accumulate PHAs were amply spread over all of the biplot space. *Ferruginibacter*, *Oscillochloris* and *Roseiflexus* dominated in the early stages of the experiment for both SBRs (NS and SS periods); *Acidovorax*, *Alkaliphilus*, *Comamonas*, *Devosia*, *Gemmobacter*, *Leifsonia*, *Lysobacter*, *Polynucleobacter*, and *Thermomonas* in NI; *Azospirillum*, *Niveispirillum* and *Pandoraea* in SI; *Chryseobacterium* and *Acinetobacter* in NII, *Calothrix*, *Dyella*, *Flavobacterium*, and *Novosphingobium* in NIII and SII; and finally, *Corynebacterium*, *Dokdonella* and *Qipengyuania* in SIII. The remaining genera (*Gordonia*, *Pseudomonas*, *Rhizobium*, *Stenotrophomonas*, *Taibaiella*, and *Tsukamurella*) were not linked to any specific experimental period. Moreover, high PHAs production performances were achieved in both SBRs, confirming the plasticity and versatility of their dominant bacterial communities to fast-changing conditions. In this sense, Huang et al. [17] have previously described strong dynamic successions of bacterial communities during operational periods in different SBRs while the functional stability was maintained, in terms of the PHAs accumulation capacity. However, Wen et al. [38] recently described that the maximum PHAs yield of an MMC fed with crude glycerol was found when the bacterial community was highly stable. Hence, the different strategies applied in the SBRs strongly modulated the PHA-accumulating bacterial community towards more efficient polymer accumulation yields in coping with the changes in operational parameters or other fluctuating situations [124]. Finally, the NaCl concentration of 10 g/L had no negative effect on the establishment of an effective PHA-accumulating community, supporting that salinity conditions do not inhibit PHAs accumulation. Therefore, there was a functional redundancy within the mixed microbial community of both inocula, and the different enrichment strategies applied thus selected the most adapted bacteria to each salinity condition.

In addition, several strong correlations were found among the abiotic and the biotic parameters, according to Pearson’s product–moment correlation coefficients (Appendix A). The pH of the feeding and the PHAs accumulation yield were the variables displaying more robust correlations with the changes in the bacterial community structure. Furthermore, the pH of the feeding was controlled by an NHCO_3_ addition, which was reduced in the NIII and SIII periods, leading to a substantial impact on the structure of the bacterial community, but without hampering PHAs accumulation yields. In fact, higher PHAs accumulation yields were linked to the bacterial community structures of periods NII, NIII, SII, and SIII. These results confirmed that the implementation of the DGL strategy was successful in the enrichment of PHA-accumulating genera, regardless of pH control. Moreover, it should be pointed out that the PHAs yield was positively correlated with higher consumption of TN (*r* = 0.96) and COD (*r* = 0.95), and with the pH of the feeding (*r* = 0.80), but negatively correlated with pH at the end of the cycle (*r* = −0.89) and the VSS (*r* = −0.96). In this sense, Oliveira et al. [41] indicated that, although PHAs accumulation was traditionally linked to low organic rate loads to effectively reduce nutrient availability during the famine phase, a higher supply of nutrients would allow higher biomass concentration and biomass production, increasing the PHAs accumulation capacity. Therefore, the positive effect of higher consumption of TN and COD on the PHA-accumulating capacity of the SBRs suggests an increased bacterial metabolism during the feast phase that raised the reactors’ PHAs yield.

Regarding the relationships between the PHAs yields and the RAs of the PHA-accumulating genera, higher RAs of *Acinetobacter*, *Calothrix*, *Dyella*, *Novosphingobium*, *Qipengyuania*, and *Tsukamurella* were related to higher PHAs concentrations within the reactors. These genera stand out as key PHA-accumulating bacteria when using a DGL strategy, regardless of the acid conditions, as they were linked to periods NII, NIII, SII, and SIII. However, several PHA-accumulating genera were unexpectedly negatively linked to the PHA-accumulating capacity, as are *Alkaliphilus*, *Comamonas*, *Chryseobacterium*, *Gemmobacter*, *Lysobacter*, *Oscillochloris*, *Polynucleobacter*, *Pseudomonas*, *Roseiflexus*, *Stenotrophomonas*, and *Thermomonas*. On the other hand, there were higher RAs of the minority genera that were negatively correlated with the PHAs yields. These results agreed with Huang et al. [17], which previously determined that higher PHA-accumulating abilities were found when fewer members were present in the bacterial community.

To sum up, the different selective pressures applied resulted in strong successions of the bacterial communities leading to an enrichment of PHA-accumulating genera and thus improved the PHAs yields in both SBRs, mainly under the DGL strategy, and regardless of the pH of the feeding. Moreover, despite the differences in the structure of the bacterial communities between SBR-N and SBR-S, there were no significant differences in their PHA-accumulating yields. Therefore, neither the source of inoculum nor the presence of NaCl had negative impacts on establishing effective PHA-accumulating communities in SBRs fed with industrial waste fish oil. However, the average accumulation yields achieved in both SBRs (7.93% and 8.56% for NIII and SIII, respectively) and the maximum PHAs accumulation capacities displayed (22.03% and 26.33% at days 264 and 63 for SBR-N and SBR-S, respectively) were lower than those described in other studies using pure cultures and specific substrates [125,126]. Therefore, further research is necessary to optimize the operational parameters using a DGL strategy to improve the PHAs accumulation performance of MMCs in lab-scale SBRs fed with residual FOG-enriched streams, before scaling up the process.

## 4. Conclusions

The results of the present study provided valuable knowledge regarding the PHA-storing bacterial communities in SBRs using FOG-enriched fish-canning waste as the substrate. Highly dynamic changes in the structure of the bacterial communities were observed in the MMCs of the two SBRs operated under either non-saline or saline conditions. A higher level of NaCl (10 g/L) was not an obstacle for an adequate enrichment of PHA-accumulating genera in the MMC. The DGL strategy was revealed as a successful tool to drive the modulation of the community structure of the MMC towards an increase in the accumulation of PHAs. The genera *Acinetobacter*, *Calothrix*, *Dyella, Novosphingobium*, *Qipengyuania*, and *Tsukamurella* were those more strongly correlated with PHAs accumulation. Therefore, they are proposed as promising genera with the ability to accumulate PHAs in MMC using fish-canning wastes.

The findings described in this paper helped to understand the contribution of the operating conditions to the shaping of successful PHA-accumulating MMCs, and provided the identification of the key genera selected under different salinities. This knowledge sets the basis for the design of improved full-scale processes to valorize fish-canning wastes into bioplastics.

## Figures and Tables

**Figure 1 polymers-14-01396-f001:**
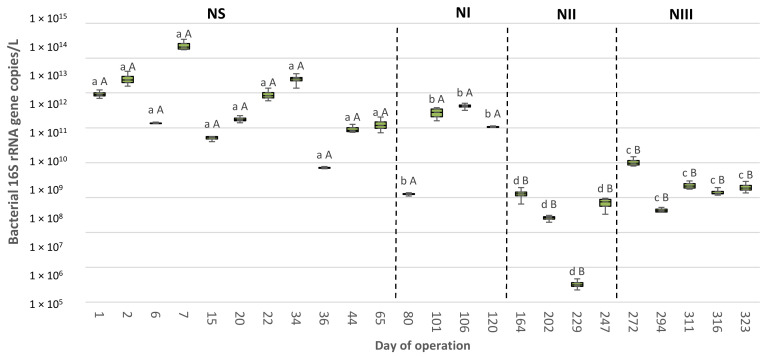
Bacterial 16S rRNA genes copies per liter of enriched activated sludge determined by quantitative PCR in sludge samples (*n* = 2) retrieved from an enrichment SBR operated under non-saline conditions (SBR-N, 0.5 g NaCl/L). Different lowercase letters indicate significant differences among periods for a given bioreactor (NS, NI, NII, or NIII; SS, SI, SII, or SIII) according to the Kruskal–Wallis and Conover–Iman tests (*p* < 0.05). Different capital letters indicate significant pairwise differences between bioreactors for a given period (NS vs. SS; NI vs. SI; NII vs. SII; or NIII vs. SIII) according to the Mann–Whitney test (*p* < 0.05).

**Figure 2 polymers-14-01396-f002:**
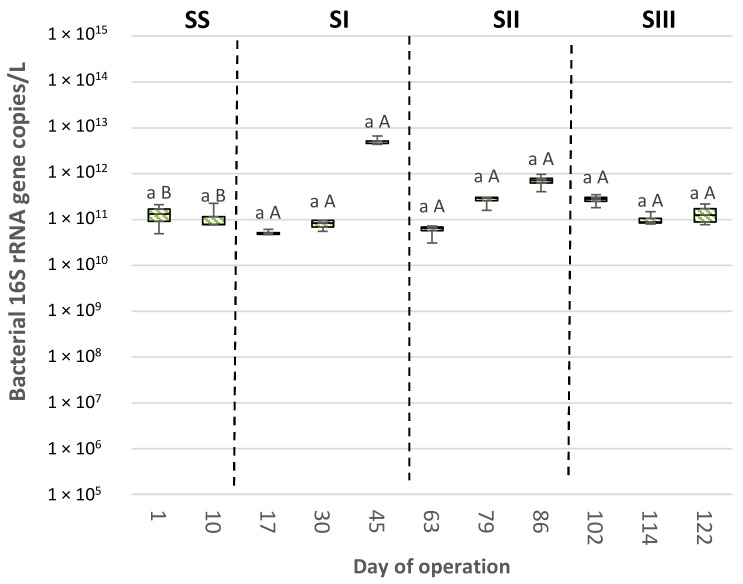
Bacterial 16S rRNA genes copies per liter of enriched activated sludge determined by quantitative PCR in sludge samples (*n* = 2) retrieved from an enrichment SBR operated under saline conditions (SBR-S, 10 g NaCl/L). Different lowercase letters indicate significant differences among periods for a given bioreactor (NS, NI, NII, or NIII; SS, SI, SII, or SIII) according to the Kruskal–Wallis and Conover–Iman tests (*p* < 0.05). Different capital letters indicate significant pairwise differences between bioreactors for a given period (NS vs. SS; NI vs. SI; NII vs. SII; or NIII vs. SIII) according to the Mann–Whitney test (*p* < 0.05).

**Figure 3 polymers-14-01396-f003:**
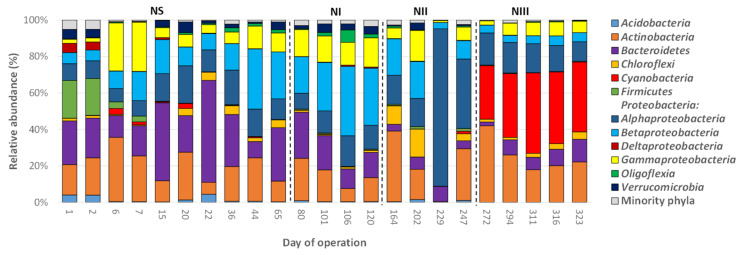
Average relative abundance of the main bacterial phyla (RA > 0.5%) found in biomass samples from an enrichment SBR operated under non-saline conditions (SBR-N, 0.5 g NaCl/L).

**Figure 4 polymers-14-01396-f004:**
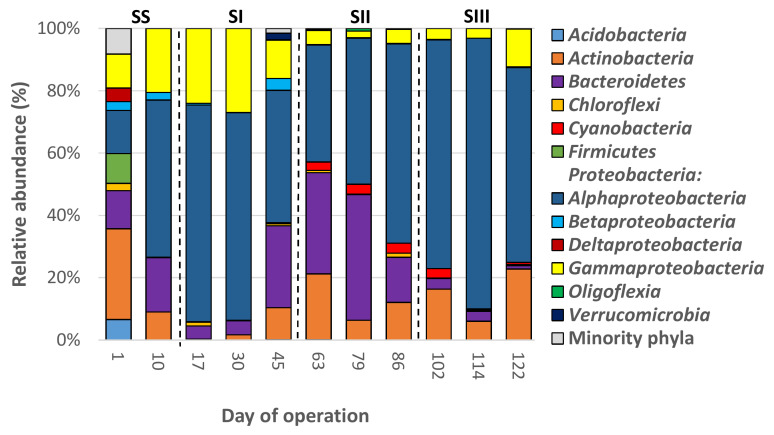
Average relative abundance of the main bacterial phyla (RA > 0.5%) found in biomass samples from an enrichment SBR operated under saline conditions (SBR-S, 10 g NaCl/L).

**Figure 5 polymers-14-01396-f005:**
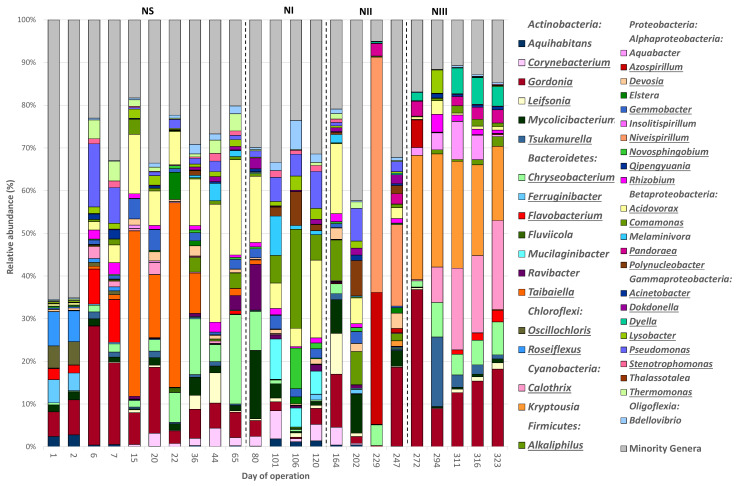
Average relative abundance of the main bacterial genera (RA > 0.5%) found in biomass samples retrieved from an enrichment SBR operated under non-saline conditions (SBR-N, 0.5 g NaCl/L). Genera potentially able to accumulate PHAs or described abundant in other PHA-accumulating SBRs are underlined.

**Figure 6 polymers-14-01396-f006:**
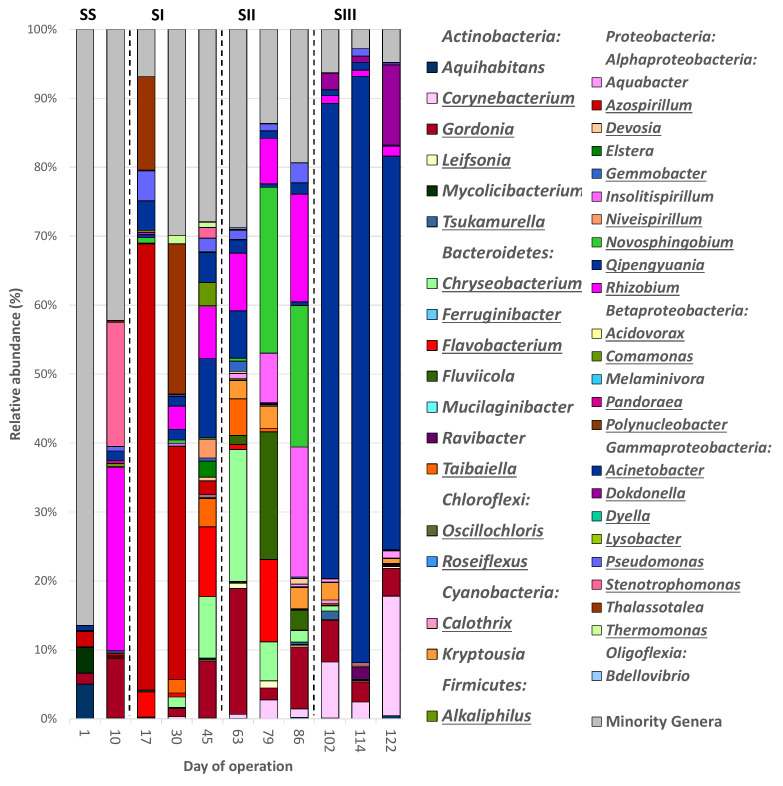
Average relative abundance of the main bacterial genera (RA > 0.5%) found in biomass samples retrieved from an enrichment SBR operated under saline conditions (SBR-S, 10 g NaCl/L). Genera potentially able to accumulate PHAs or described abundant in other PHA-accumulating SBRs are underlined.

**Figure 7 polymers-14-01396-f007:**
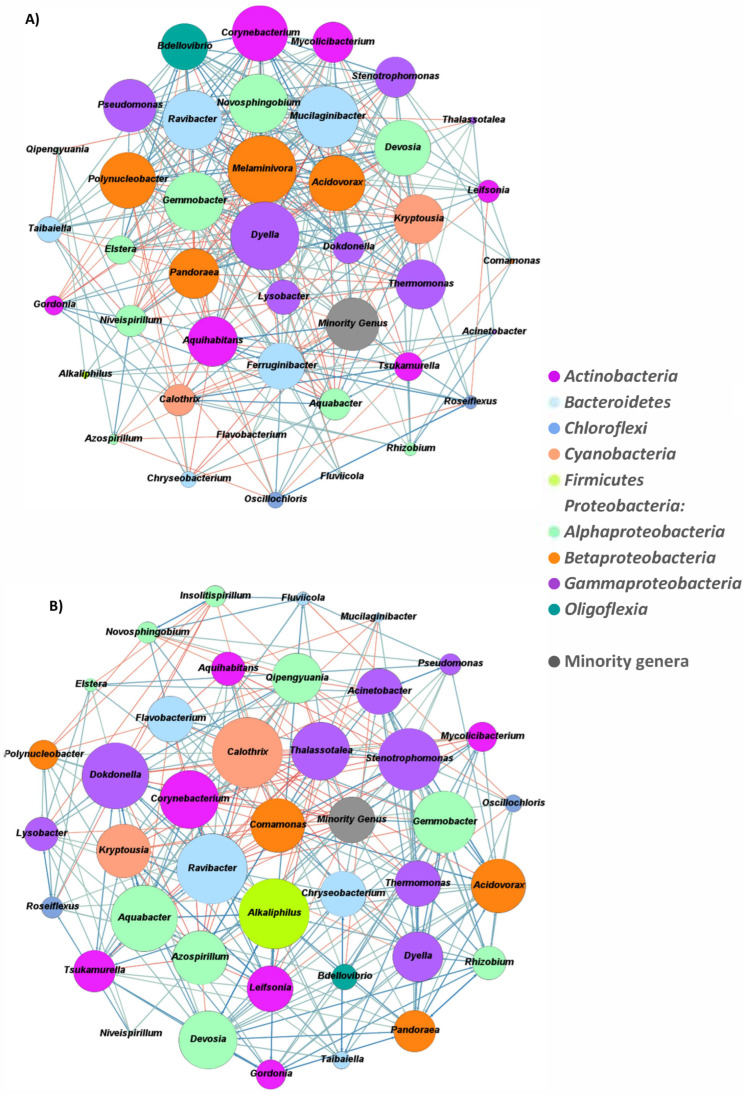
Co-occurrence networks of the significant Spearman’s rank correlation coefficients (ρ > 0.4) of the RAs of the dominant bacterial genera in biomass samples retrieved from two PHA-accumulating SBR reactors operated under (**A**) non-saline (SBR-N) and (**B**) saline (SBR-S) conditions.

**Figure 8 polymers-14-01396-f008:**
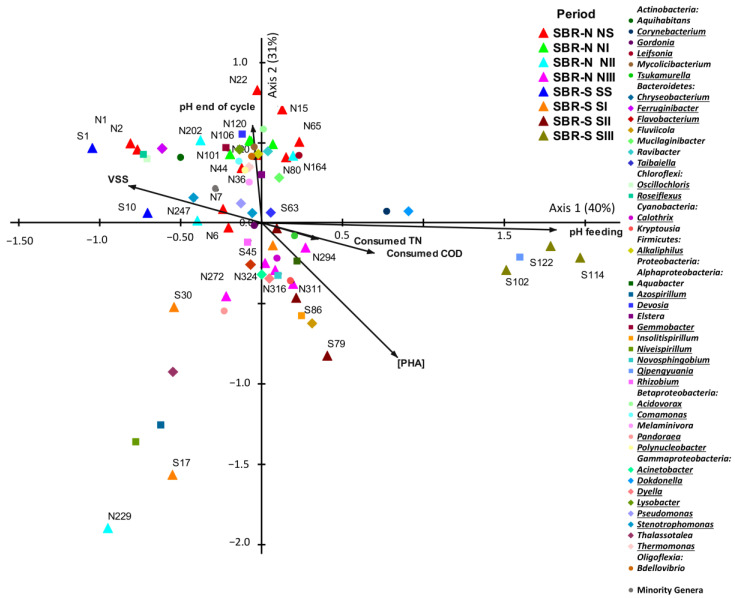
Nonmetric multidimensional scaling (NMS) ordination of dominant bacterial genera (RA < 0.5%) found in biomass samples retrieved from two SBRs operated under non-saline (SBR-N, 0.5 g NaCl/L) and saline (SBR-S, 10 g NaCl/L) conditions, and their links with the operational variables and PHAs accumulation yields. [PHA] (PHAs accumulation yield), pH feeding (pH of the feeding stream), Consumed COD (Total COD consumed), Consumed TN (Total N consumed), VSS (volatile suspended solids concentration in the reaction medium), pH end of cycle (pH at the end of the cycle). Genera potentially able to accumulate PHAs or described abundant in other PHA-accumulating SBRs are underlined.

## Data Availability

The data presented in this study are available on request from the corresponding author.

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
