# Peer review of "Dynamics of PHA-Accumulating Bacterial Communities Fed with Lipid-Rich Liquid Effluents from Fish-Canning Industries"

_polymers, 2022, doi:10.3390/polym14071396_

Round 1

Reviewer 1 Report

Dear all,

Greetings

Please find enclosed my comments regarding this article

Referenced as: polymers-1645755

Titled: Assessing the dynamics of PHA-accumulating enriched mixed bacterial communities fed with lipid-rich liquid effluents generated from the fish-canning industry

 The authors have performed good, actual and promising work concerning the new technology to replace petroleum-based plastics by using PHA-accumulating bacteria, but this article can be accepted for publication in Polymers, after addressing and fixing all these points (Minor Revisions)

1) Title: very long

2) Abstract: please add in your study the PHAs accumulation yields to avoid confusion of readers and they can have a clear comparison

3) Keywords: ok but there is mixture between some abbreviations

4) Comments:

- Methods [21-26] instead of methods [21,22,23,24,25,26].

- Add the appropriate reference in the introduction near to each cited property

- Figures 1,2 and 3 replaced them by two figures is much better (they contain a lot of information and data)

- 3.4. Dynamics of the bacterial communities' structure (move it to the next page)

- Figure 4 not clear please make it bigger

- in the conclusion you can propose new perspectives

- References: ok

With regards

Author Response

1) Title: very long

The title had been changed to “Dynamics of PHA-accumulating bacterial communities fed with lipid-rich liquid effluents from the fish-canning industry”.

2) Abstract: please add in your study the PHAs accumulation yields to avoid confusion of readers and they can have a clear comparison

The change has been included in the abstract section as requested by the reviewer

3) Keywords: ok but there is mixture between some abbreviations:

The keywords were changed to avoid abbreviations

4) Comments:

- Methods [21-26] instead of methods [21,22,23,24,25,26].

The change has been included as requested by the reviewer. Also in line 92 (33,34,35) was replaced with (33-35).

- Add the appropriate reference in the introduction near to each cited property

A critical review of the introduction was made and the appropriate references were added to each property.

- Figures 1,2 and 3 replaced them by two figures is much better (they contain a lot of information and data)-

Each figure was split into two figures

- 3.4. Dynamics of the bacterial communities' structure (move it to the next page)

The final position will be adjusted by the editorial team

- Figure 4 not clear please make it bigger

The figure 4 was rebuilt according to this comment

- in the conclusion you can propose new perspectives

The new sentence has been added as new perspectives in the conclusion sectiomn “The findings described in this paper helped to understand the contribution of the operating conditions to the shaping of successful PHA-accumulating MMCs and provided the identification of the key genera selected under different salinities. This knowledge sets the basis for the design of improved full-scale processes to valorize fish-canning wastes into bioplastics”.

Reviewer 2 Report

This is a interesting article without flaws. David et al. talks about the changes in size and structure of bacteria communities supplied with industrial wastes during PHA accumulation. Authors used sequencing approach to identify the bacterial communities and it showed Acinetobacter, Calothrix, Dyella, Flavobacterium, Novosphingobium, Qipengyuania, and Tsukamurella as the  essential PHA-accumulating genera regardless of conditions used. Article was well written and the experimental methods were clearly described. 

Does the authors tried to isolate the pure culture from mixed bacterial communities and characterized?

It is interesting that high salt condition doesn't affect the bacterial population. What is the potential reason?

Author Response

Does the authors tried to isolate the pure culture from mixed bacterial communities and characterized?

The main objective of the present study was to survey the structure and dynamics of the bacterial communities enriched with the fish-canning effluent waste and identify the potential PHA-accumulating bacteria. For this purpose, we applied cultivation-independent techniques only. However, we agree with the reviewer that once the community has been characterized, it would be very interesting to attempt the isolation of the key PHA-accumulating bacterial genera. We definitely plan to try this approach in future studies.

It is interesting that high salt condition doesn't affect the bacterial population. What is the potential reason?

The structure of the PHA-accumulating mixed microbial cultures differed among the two SBRs which were operated at either 0.5 or 10 g/L NaCl, at both Phyla and Genus levels (Figs. 2 and 3). Nonetheless, we highlighted that these differences did not significantly influence the average PHA accumulating yields. We assume that this fact reflects functional redundancy within the mixed microbial community. The different PHA-accumulating genera experience shifts in their relative abundances according to the changes in the environmental variables which are more favorable to each of them, including salinity.

The sentence “Therefore, there was a functional redundancy within the mixed microbial community of both inocula, and the different enrichment strategies applied selected the most adapted bacteria to each salinity condition” was added in the new version